

# Global Modeling of Ice Nucleating Particles of Multiple Aerosol Species and Associated Cloud Radiative Effects

Kei Kawai[1], Zhaoyi Ren[1], and Hitoshi Matsui[1]

[1]Graduate School of Environmental Studies, Nagoya University, Nagoya, 464-8601, Japan

*Correspondence to*: Kei Kawai (kkawai@nagoya-u.jp) and Hitoshi Matsui (matsui@nagoya-u.jp)

**Abstract.** A subset of aerosol species act as ice nucleating particles (INPs) in mixed-phase clouds, where they influence cloud distributions and lifetimes and thus Earth's radiative balance through aerosol-cloud interactions. However, few modeling studies have simultaneously considered multiple aerosol species as INPs, and the radiative effects associated with INPs remain poorly constrained. This study uses a global climate-aerosol model to evaluate the number concentrations, spatial distributions,

and cloud radiative effects of INPs from multiple aerosol species, including dust, bioaerosols, marine organic aerosol (MOA), and black carbon. The model reproduces global INP observations more accurately when multiple INP sources are included compared to simulations that consider dust INPs alone. Dust accounts for 97 % of the global mean INP number concentration in clouds because of its large atmospheric abundance. However, bioaerosols—particularly bacteria with high ice nucleating ability at relatively warm temperatures ($> -10$ °C)—dominate INPs in the middle troposphere at low latitudes and in the lower

troposphere at mid-latitudes in the Northern Hemisphere. MOA dominates INPs in the middle and lower troposphere at middle and high latitudes in the Southern Hemisphere, where concentrations of other INP-active aerosols are low. Incorporating observational constraints on the temperature dependence of INPs increases the global mean cloud radiative effect of total INPs from +0.071 to +0.19 W m$^{-2}$. These findings underscore the importance of including INPs from multiple aerosol species in climate models for better understanding of aerosol-cloud interactions via INPs.

## 1 Introduction

Ice nucleating particles (INPs) play a critical role in the microphysics of mixed-phase clouds, which consist of supercooled water droplets and ice crystals, by initiating formation of ice crystals at subzero temperatures (approximately $-37$ °C to 0 °C) (Kanji et al., 2017). Once ice crystals form, water droplets evaporate and transfer water vapor to the growing ice crystals via the Wegener-Bergeron-Findeisen process (Korolev, 2007), altering the phase balance within the cloud. This process decreases

cloud water content, reduces cloud albedo, and influences Earth's radiative balance through aerosol-cloud interactions (Shi & Liu, 2019; Storelvmo, 2017). Despite their importance, the radiative effects of aerosol-cloud interactions via INPs remain highly uncertain, primarily because of the complex and diverse nature of INP sources and their interactions with clouds (Murray et al., 2021).





Some aerosol species act as INPs in mixed-phase clouds, facilitating ice crystal formation at different subzero temperatures
and across various spatial distributions (Kanji et al., 2017; Murray et al., 2012). Mineral dust emitted from arid and semi-arid
regions such as the Sahara and Gobi deserts is recognized as one of the most important sources of INPs because of its
atmospheric abundance and high ice nucleating ability at temperatures colder than approximately −15 °C (Atkinson et al.,
2013; DeMott et al., 2015). Recent studies have also shown that Arctic dust that originates from snow- and vegetation-free
surfaces in the Arctic region has significantly higher ice nucleating ability than desert dust, especially at temperatures warmer
than −20 °C (Tobo et al., 2019), and it plays a dominant role in dust INPs in the lower troposphere of the Arctic during summer
and fall (Kawai et al., 2023; Matsui et al., 2024).

Bioaerosols (or primary biological aerosol particles), including bacteria, fungal spores, and pollen, are another important
source of INPs. In particular, some bacterial species such as *Pseudomonas syringae* nucleate ice crystals efficiently at relatively
warm subzero temperatures (> −10 °C) (Diehl & Mitra, 2015; Hummel et al., 2018). Marine organic aerosol (MOA), produced
through oceanic biological processes and bubble-bursting mechanisms, serves as an additional source of INPs, particularly in
remote marine and polar regions (Burrows et al., 2013; Wilson et al., 2015). Black carbon (BC), a product of incomplete
combustion, has also been identified as a potential source of INPs, although its ice nucleating ability is lower than that of other
aerosol species (Bond et al., 2013; Kanji et al., 2017). Despite the diversity of INP sources, few modeling studies have
simultaneously considered multiple aerosol species as INPs, and their combined cloud radiative effects (CREs) remain poorly
understood. In a recent modeling study by Chatziparaschos et al. (2024), multiple aerosol species were considered as INPs,
but their CREs were not quantified. To improve our understanding of aerosol-cloud interactions, which contribute to one of
the largest sources of uncertainty in climate change projections, it is necessary to develop global climate models that
incorporate diverse sources of INPs and evaluate their relative contributions to concentrations of INPs and their associated
CREs on a global scale.

In this study, we evaluate the contributions of various species of aerosols (dust, bioaerosols, MOA, and BC) to global and
regional INP number concentrations and their CREs via INPs using a global climate-aerosol model. We also estimate the range
of CREs of all aerosol species via INPs by incorporating a new observational constraint on ice nucleating ability. The findings
highlight the importance of accounting for multiple INP sources in global climate models to reduce uncertainties in estimating
concentrations of INPs and their CREs and to improve our understanding of aerosol-cloud interactions.

## 2 Methods

### 2.1 Global climate-aerosol model

We used the Community Atmosphere Model (CAM) version 5 (Neale et al., 2012; Lamarque et al., 2012) with the Aerosol
Two-dimensional bin module for foRmation and Aging Simulation (ATRAS) version 2 (Matsui, 2017; Matsui & Mahowald,
2017), and the Community Land Model (CLM) version 4 (Oleson et al., 2010) within the framework of the Community Earth
System Model version (CESM) 1.2.0 (Hurrell et al., 2013). The CAM-ATRAS model considers seven aerosol species (dust,



sea salt, black carbon, sulfate, nitrate, ammonium, and organic aerosol) along with the bioaerosols and MOA added for this study. Aerosol particles with dry diameters ranging from 1 nm to 10 μm are represented across 12 size bins, except for bioaerosols and MOA. The model simulates key aerosol processes, including emissions; new particle formation; condensation of sulfate, nitrate, and organic aerosols; coagulation; aqueous-phase chemistry; dry and wet deposition; cloud activation; and

aerosol-radiation and aerosol-cloud interactions.

The extensive evaluation of the model against various aerosol observations in our previous studies has demonstrated its reliability in reproducing aerosol concentrations, size distributions, and optical properties (Kawai, Matsui, Kimura, & Shinoda, 2021; Kawai, Matsui, & Tobo, 2021; Kawai et al., 2023; Liu et al., 2022; Liu, Matsui, et al., 2024; Liu, Song, et al., 2024; Matsui & Mahowald, 2017; Matsui, Hamilton, & Mahowald, 2018; Matsui, Mahowald, et al., 2018; Matsui et al., 2024).

In this study, the model considers dust (Arctic and non-Arctic dust), bioaerosols (bacteria, fungal spores, and pollen), MOA, and BC (from biomass burning) as INPs (Table 1). The emissions and ice nucleating abilities of each species are described below.

**Table 1.** Aerosol species and types acting as INPs in this study, and data sources for their emissions and ice nucleating abilities.

| Aerosol species | Aerosol type | Emission | Ice nucleating ability |
|---|---|---|---|
| Dust | Arctic dust | Online (Zender et al., 2003; Kok et al., 2014) | Tobo et al. (2019); Kawai et al. (2023) |
| | Non-Arctic dust | | DeMott et al. (2015) |
| Bioaerosols | Bacteria | Offline (Hoose et al., 2010) | Diehl and Mitra (2015) |
| | Fungal spores | | Hummel et al. (2018) |
| | Pollen | | Diehl and Mitra (2015) |
| MOA | MOA | Online (Burrows et al., 2013; Wilson et al., 2015) | Wilson et al. (2015) |
| BC | Biomass burning BC | Dataset (van Marle et al., 2017) | Schill et al. (2020) |




### 2.1.1 Dust

Dust emission fluxes are calculated online based on wind speed and land surface conditions, including soil moisture, vegetation, and snow cover, following the schemes of Zender et al. (2003) and Kok et al. (2014). The size distribution of emitted dust particles is based on Kok (2011). Dust emitted north and south of 60° N is defined as Arctic dust and non-Arctic dust, respectively. Transport, deposition, and ice nucleation processes for Arctic and non-Arctic dust are calculated independently using separate tracers (Kawai et al., 2023).

The ice nucleating abilities of Arctic and non-Arctic dust for immersion and condensation freezing are parameterized following Tobo et al. (2019) and DeMott et al. (2015), respectively. Observations by Tobo et al. (2019) have demonstrated that Arctic dust has significantly higher ice nucleating ability than desert dust, particularly at temperatures warmer than −20 °C (see Kawai et al. (2023) for the equation). DeMott et al. (2015) have measured the ice nucleating ability of desert dust using field sampling and aircraft observations of Saharan and Asian dust.

Using these parameterizations, dust INP number concentrations are calculated online as a function of air temperature (between −37 °C and −5 °C) and dust mass or number concentrations (both interstitial-phase and cloud-phase) in the presence of liquid water droplets. In this study, we have reported the number concentrations of INPs within clouds, which were equated to the grid-mean INP number concentrations multiplied by the fractions of stratus clouds. This calculation is consistent with the CAM5 cloud microphysics scheme (Gettelman et al., 2010; Morrison & Gettelman, 2008). Although the INP parameterizations differ among aerosol species, INP number concentrations from bioaerosols, MOA, and BC are calculated using the same computational approach. The total dust INP number concentrations are equated to the sum of Arctic and non-Arctic dust INP number concentrations.

The model has been demonstrated to reasonably reproduce global satellite dust observations (Kawai et al., 2023), surface dust observations across various regions (Kawai, Matsui, Kimura, & Shinoda, 2021; Matsui & Mahowald, 2017; Matsui et al., 2024), ice-core dust deposition observations in the Arctic (Matsui et al., 2024), $PM_{10}$ observations in East Asia, and INP observations in Tokyo (Kawai, Matsui, & Tobo, 2021) and the Arctic (Kawai et al., 2023).

### 2.1.2 Bioaerosols

The model in this study newly incorporates three types of bioaerosols: bacteria, fungal spores, and pollen. Their emission fluxes are calculated based on terrestrial vegetation and meteorological conditions following the scheme of Hoose et al. (2010). Bioaerosol emission fluxes for the year 2010 are calculated in this study and provided as input to the model. The sizes of emitted particles are assumed to be 1 μm for bacteria, 5 μm for fungal spores, and 30 μm for pollen.

The ice nucleating abilities of 4 % of bacteria (*Pseudomonas syringae*) and 100 % of pollen (tree pollen) are based on Diehl and Mitra (2015). We assumed that 29 % of *Cladosporium* sp. fungal spores and 8 % of *Mortierella alpina* fungal spores act as INPs following Hummel et al. (2018). The bioaerosol INP number concentrations are equated to the sum of the INP number concentrations of bacteria, fungal spores, and pollen.



### 2.1.3 MOA

The model also newly incorporates MOA in this study, and its emission fluxes are calculated based on oceanic biogeochemistry
and sea salt emission following the methods of Burrows et al. (2013) and Wilson et al. (2015). Climatological monthly data of
particulate organic carbon concentrations in seawater, retrieved from the Moderate Resolution Imaging Spectroradiometer
(MODIS) onboard the Aqua satellite during 2002–2023, are used for the calculation. Missing values at high latitudes are
imputed using the average north of 30° N for the Northern Hemisphere and south of 30° S for the Southern Hemisphere. The
emitted MOA particles are assumed to have a diameter of 0.2 μm (Burrows et al., 2013). The ice nucleating ability of MOA is
parameterized as a function of air temperature and organic content based on Wilson et al. (2015).

### 2.1.4 BC

The model considers two types of BC sources: anthropogenic and biomass burning emissions. BC emission data are provided
from the Coupled Model Intercomparison Project phase 6 (CMIP6) emission dataset for the year 2010 (Hoesly et al., 2018;
van Marle et al., 2017). The ice nucleating ability of biomass burning BC is based on the parameterization of Schill et al.
(2020). Because anthropogenic BC does not act efficiently as INPs under mixed-phase cloud conditions (Kanji et al., 2017),
this study does not consider its contribution as INPs.

### 2.2 Base simulation

We conducted the Base simulation for 11 years (2009–2019) using a horizontal resolution of 1.9° × 2.5° and 30 vertical layers
extending from the surface to an altitude of approximately 40 km. Monthly sea surface temperature and sea ice concentration
data were prescribed from Hurrell et al. (2008). The first year of the simulation (2009) was treated as model spin-up, and the
last 10 years (2010–2019) were used for analysis. Temperature and horizontal wind fields in the free troposphere (<800 hPa)
were nudged to the Modern-Era Retrospective analysis for Research and Applications version 2 (MERRA2) dataset. The
CMIP6 dataset for the year 2010 (Hoesly et al., 2018; van Marle et al., 2017) was used for anthropogenic and biomass burning
emissions of aerosols and their precursors. Dust and sea salt emissions were calculated online within the model (Kok et al.,
2014; Mårtensson et al., 2003).

### 2.3 Observationally constrained simulation

A sensitivity simulation (the observationally constrained simulation) was conducted in which the ice nucleating ability of
bacteria was adjusted to match the temperature dependence of the annual mean INP number concentrations observed in Tokyo
from August 2016 to July 2017 (Tobo et al., 2020) (Fig. S1). The ice nucleating ability of bacteria was multiplied by $10^{3.2}$ at
$\leq -35$ °C, $10^{0.2}$ at $-19$ °C, $10^{-0.3}$ at $-14$ °C, $10^{-1.8}$ at $-9$ °C, and $10^{1.8}$ at $-5$ °C, with a lognormal interpolation between these
temperatures. In Sect. 3.4, this simulation is used to estimate the CREs of total INPs and thus to provide a range of CRE
estimates when compared to the Base simulation. For simplicity, the ice nucleating ability of bacteria was adjusted to align



with the observations in this simulation, but this adjustment does not imply that the ice nucleating ability of bacteria is the only source of uncertainty. This adjustment includes uncertainties related to the spatial distributions and ice nucleating abilities of

all aerosol species as well as the potential influence of unknown INP sources.

## 2.4 CREs via INPs

We also conducted sensitivity simulations in which the number concentrations of dust, bioaerosol, or total INPs were increased by a factor of 10 to investigate their CREs. The CREs via INPs were calculated as the difference in cloud radiative forcing between the 10-fold and 1-fold simulations. Cloud radiative forcing was computed as the difference in radiative fluxes at the

top of the atmosphere between all-sky and clear-sky conditions (Ramanathan et al., 1989). The 10-fold sensitivity simulations were performed to account for current uncertainties in the ice nucleating abilities of the aerosol species and to obtain a high signal-to-noise ratio (Kawai, Matsui, & Tobo, 2021; Kawai et al., 2023; Shi & Liu, 2019). The estimated CREs can therefore be interpreted as values close to the upper limit of the CREs via INPs.

We further estimated the scaled net CREs corresponding to the presence of INPs (1-fold − 0-fold) by fitting the simulated

net (shortwave + longwave) CREs for 1–50-fold dust INPs relative to 0-fold dust INPs using a power-law function of $y = ax^b$ (Fig. S2). The best-fit parameters were $a$ = 0.0576 and $b$ = 0.640, yielding a scaling factor of 0.297 to convert the CREs estimated from the 10-fold − 1-fold simulations to those corresponding to the 1-fold − 0-fold differences. This fitting approach provides a higher signal-to-noise ratio for the purpose of estimating the INP-induced CREs than simply taking the difference between the 1-fold and 0-fold simulations.

## 2.5 Observational data

To evaluate simulated aerosol and INP concentrations, bioaerosol observations in the Amazon (Patade et al., 2021) and global INP observations (Hu et al., 2023; Mason et al., 2016; Tobo et al., 2019, 2020) were used. For comparisons with the INP observations, simulated INP number concentrations and their temperature dependence were calculated offline using observed temperatures and simulated daily mean aerosol concentrations during each observation period. Stratus cloud fractions (Sect.

2.1.1) were not considered in these offline INP calculations.

## 3 Results and discussion

## 3.1 Comparisons with INP observations

We compare our model estimates with INP observations from various locations worldwide (Fig. 1). When the model considers only dust as the source of INPs (red in Fig. 1a–1d), it underestimates the observed INP number concentrations by 1–4 orders

of magnitude in China, Canada, France, and the Arctic. However, the model has a better agreement with the observations when bioaerosols, MOA, and BC are incorporated as additional sources of INPs (green and blue in Fig. 1a–1d). For the observations



in China (−15 °C and −8 °C), Canada, and France, bioaerosols account for the largest fraction of total INPs in the model estimates (Fig. 1e–1g), whereas MOA accounts for the largest fraction for the Arctic observations in March (−25 °C and −20 °C) (Fig. 1h). The model performance therefore improves markedly in regions where bioaerosols and MOA contribute notably to the total INPs and where dust alone cannot explain the INP observations.

We next compare the model estimates with the year-round INP observations in Tokyo (Tobo et al., 2020) (Fig. 2). When only dust is considered as the source of INPs in the model (red in Fig. 2), the model reproduces the observations reasonably well during spring and autumn, when dust particles are transported from the Gobi and Taklamakan Deserts. However, during summer and winter, the model underestimates the observed INP number concentrations by 2–3 orders of magnitude. In contrast, when bioaerosols, MOA, and BC are added as INP sources (green and blue in Fig. 2), the model successfully reproduces the observations at −15 °C and −20 °C throughout the year. At −25 °C, the model still underestimates the observations by about one order of magnitude during summer and winter. This discrepancy might be attributable to uncertainties in dust emission and transport or to unknown local sources of INPs (e.g., local soil dust). These results demonstrate that including multiple INP sources improves the model performance, particularly in seasons when dust alone cannot explain the observed INP concentrations.

The model reasonably well reproduces bioaerosol number concentrations observed in the Amazon rainforest, where bioaerosols are the dominant aerosols (Patade et al., 2021) (Fig. S3). Although the model underestimates the number concentrations of bacteria by more than one order of magnitude, it reproduces the observed number concentrations of fungal spores and pollen within one order of magnitude.



**Figure 1.** (a–d) INP number concentrations observed at (a) Beijing, China (Hu et al., 2023); (b) Vancouver Island, Canada; (c) Saclay, France (Mason et al., 2016); and (d) Svalbard, Arctic (March 2017) (Tobo et al., 2019) at different freezing temperatures (black), and those simulated for the corresponding locations and periods for dust only (red), dust + bioaerosols (green), and all INP sources (blue). Dots and bars represent the averages and ranges (where available) during the observation periods, respectively. (e–h) Fractions of dust (orange), bioaerosols (green), MOA (blue), and BC (gray) in the simulated total INP number concentrations for each location.



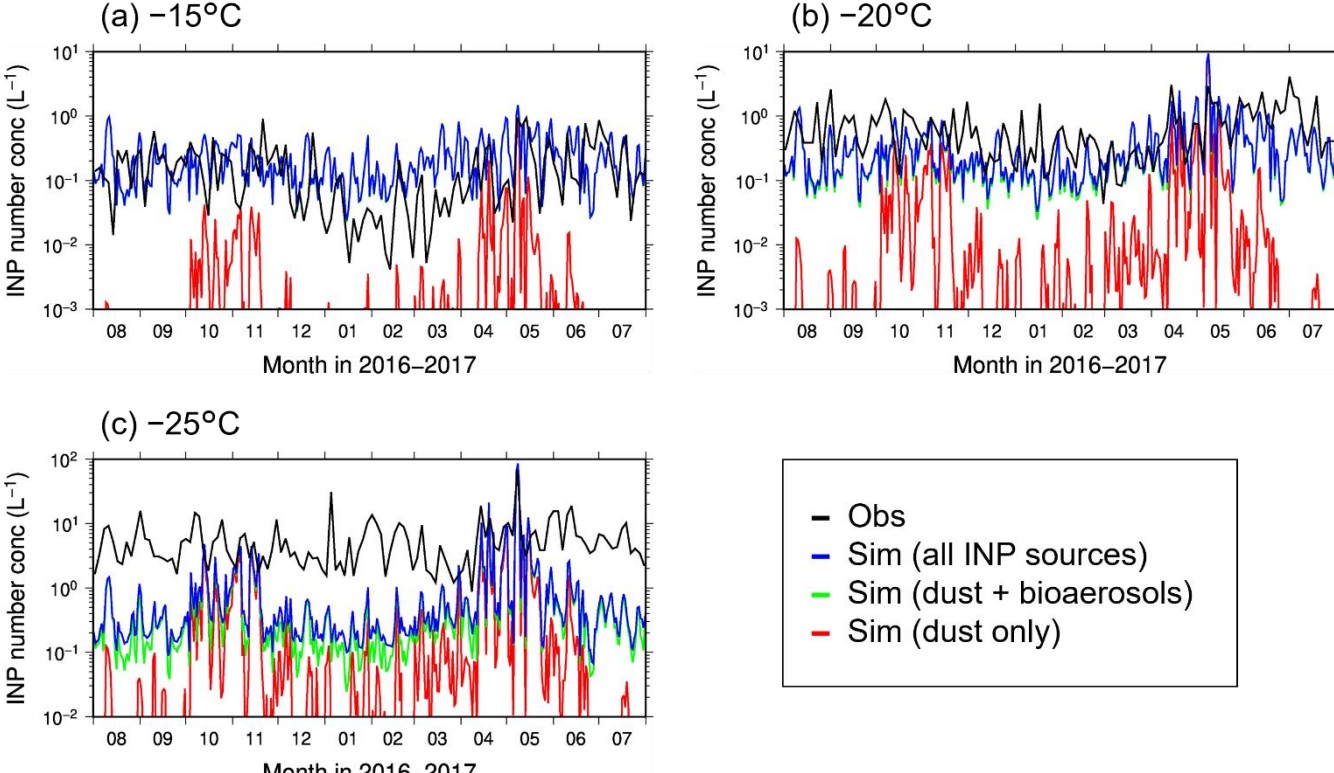

**Figure 2.** INP number concentrations observed at Tokyo Skytree, Japan, throughout the year (three-day mean) at freezing temperatures of (a) −15 °C, (b) −20 °C, and (c) −25 °C (Tobo et al., 2020) (black), and those simulated for the corresponding location and period (daily mean) for dust only (red), dust + bioaerosols (green), and all INP sources (blue).

### 3.2 Global mean INP number concentrations

We estimate the contribution of each aerosol species to the global mean INP number concentrations in clouds (Fig. 3). In the whole atmosphere, dust accounts for the largest fraction of total INPs (97 %), followed by bioaerosols (2.4 %), MOA (0.52 %), and BC (0.11 %) (Fig. 3a). In the estimates of Hoose et al. (2010) (PBAP-MAX), dust also accounts for the largest fraction of total INPs (87 %), but the fraction of BC (12 %) is larger than that of bioaerosols (0.6 %). This discrepancy is probably due to differences in the simulation periods and the ice nucleating abilities of BC used in the models.

We estimate the global mean INP number concentration per unit mass concentration for each aerosol species (Table S1), which represents their efficiency in acting as INPs. The INP-to-mass ratio is highest for MOA ($8.8 \times 10^6$ g$^{-1}$) and lowest for BC ($6.7 \times 10^5$ g$^{-1}$). Dust and bioaerosols have similar INP-to-mass ratios ($2.0 \times 10^6$ g$^{-1}$ and $2.4 \times 10^6$ g$^{-1}$, respectively), suggesting that their efficiencies are comparable as INPs. This result suggests that dust contributes the most to total INPs, primarily because it has the highest mass concentration.



In the upper and middle troposphere, dust accounts for more than 90 % of the total INP number concentrations (Fig. 3b),
primarily because of its high ice nucleating ability at lower temperatures. In the lower troposphere, dust also contributes the
largest fraction (60 %), though this fraction is smaller than its contribution in the upper and middle troposphere. The fraction
of bioaerosols is markedly higher in the lower troposphere (37 %) than in the upper (1.7 %) and middle troposphere (3.1 %)
because bioaerosols, particularly bacteria, exhibit high ice nucleating abilities at higher temperatures. However, the total INP
number concentration was smaller in the lower troposphere than in the upper and middle troposphere (Fig. 3b) mainly because
of the exponentially higher ice nucleating abilities of aerosol species at lower temperatures. The fraction of bioaerosols in the
whole atmosphere is therefore small (Fig. 3a). These results indicate that although dust is the dominant source of INPs
throughout the atmosphere, bioaerosols also play an important role in the lower troposphere.

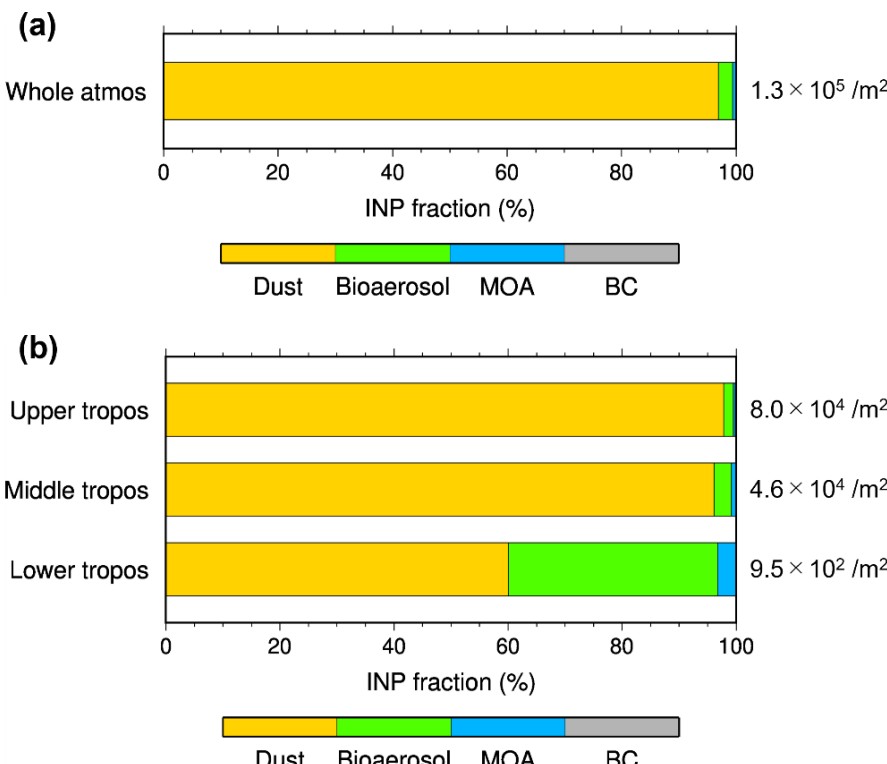

**Figure 3.** Fractions of dust (orange), bioaerosols (green), MOA (blue), and BC (gray) in the annual and global mean vertically integrated
total INP number concentrations (shown on the right) in (a) the whole atmosphere (2–1000 hPa) and (b) the upper (100–400 hPa), middle
(400–700 hPa), and lower troposphere (700–1000 hPa).



### 3.3 Global distributions of INPs

The global distribution of INPs from each aerosol species is determined by the ice nucleating ability and concentrations of the aerosol species along with ambient temperature and cloud water content. Each aerosol species is distributed over its source regions and their respective downwind areas (Fig. S4). The INPs from dust are distributed over the dust source regions, which are located on all continents except Antarctica (Fig. S5), and their downwind regions (Fig. 4a). The INPs from bioaerosols are primarily distributed over terrestrial regions globally, except Antarctica and Greenland (Fig. 4b). Among the bioaerosol types,

bacteria contribute the largest fraction of the global mean bioaerosol INP number concentration (Fig. S6). The INPs from MOA are distributed primarily over the North Atlantic to the Arctic Ocean, the North Pacific, and the Southern Ocean (Fig. 4c). The INPs from BC (from biomass burning) are abundant over South America (Fig. 4d).

We investigate the dominant aerosol species contributing the largest fraction of the vertically integrated total INP number concentration in each grid (Fig. 5a and 5b). Around Antarctica, MOA accounts for the largest fraction of total INPs, likely

because the MOA source region (Southern Ocean) is closer to Antarctica than the dust source regions (Australia, South America, and southern Africa). In Southeast Asia and central Brazil, bioaerosols account for the largest fraction of total INPs. Over the other extensive regions, dust is the dominant contributor to total INPs.

A comparison of INP number concentrations derived from dust only and from all aerosol species shows that including non-dust INP sources increases INP number concentrations by more than threefold in Antarctica, the Southern Ocean, Southeast

Asia, South Africa, and Brazil (Fig. 6). These regions correspond to areas where bioaerosols or MOA dominate total INPs (Fig. 5b). These findings highlight the importance of considering bioaerosols and MOA as sources of INPs to accurately estimate INP number concentrations in these regions.

Next, we analyze the dominant aerosol species that contribute the largest fractions of the total INP number concentrations in the upper, middle, and lower troposphere (Fig. 7). In the upper troposphere, dust is the dominant contributor across most

regions globally (Fig. 7b). In the middle troposphere, dust dominates the total INPs in the middle and high latitudes of the Northern Hemisphere, bioaerosols dominate the low latitudes of both hemispheres, and MOA dominates the middle and high latitudes of the Southern Hemisphere (Fig. 7d). Although these regions are comparable in size, the contributions of bioaerosols and MOA to the global mean total INP number concentration in the middle troposphere are much smaller than that of dust (Fig. 3b) because the INP number concentrations in regions dominated by bioaerosols and MOA are several orders of

magnitude lower than those in dust-dominated regions (Fig. 7c). In the lower troposphere, aerosols generally do not act as INPs at low latitudes because of the high temperatures ($> -5\ °C$) (Fig. 7e). In this layer, bioaerosols and dust dominate total INPs in the Northern Hemisphere, whereas MOA and bioaerosols are dominant in the Southern Hemisphere (Fig. 7f).

INPs from bioaerosols or MOA are dominant on a seasonal basis in some regions where dust INPs are dominant on an annual average (Fig. S7). For example, in the middle troposphere, bioaerosols dominate total INPs over northern Eurasia in

summer, and MOA dominates over the North Atlantic in fall and winter. In the lower troposphere over the Arctic, bioaerosol INPs are dominant in spring and MOA INPs are dominant in winter.



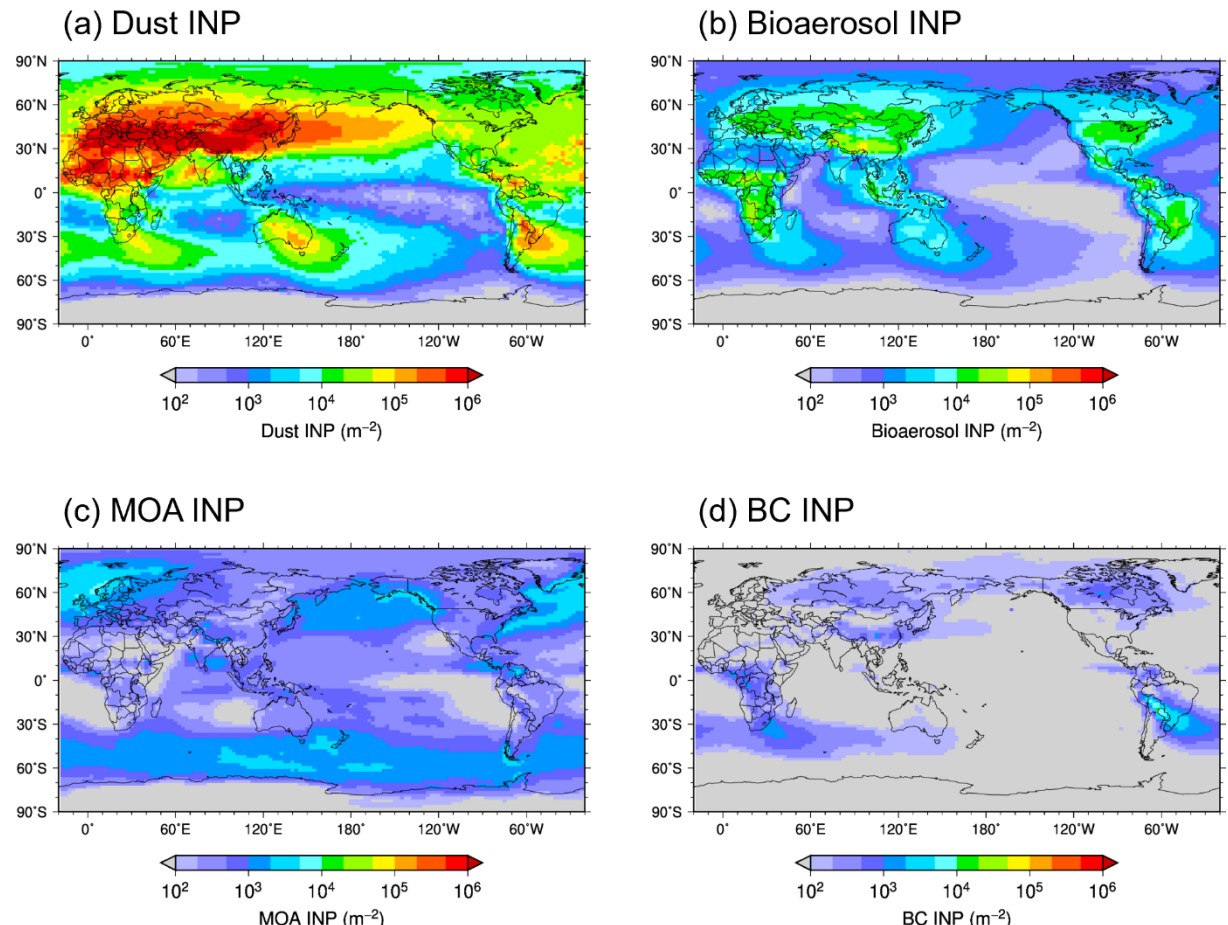

**Figure 4.** Annual mean vertically integrated INP number concentrations of (a) dust, (b) bioaerosols, (c) MOA, and (d) BC.




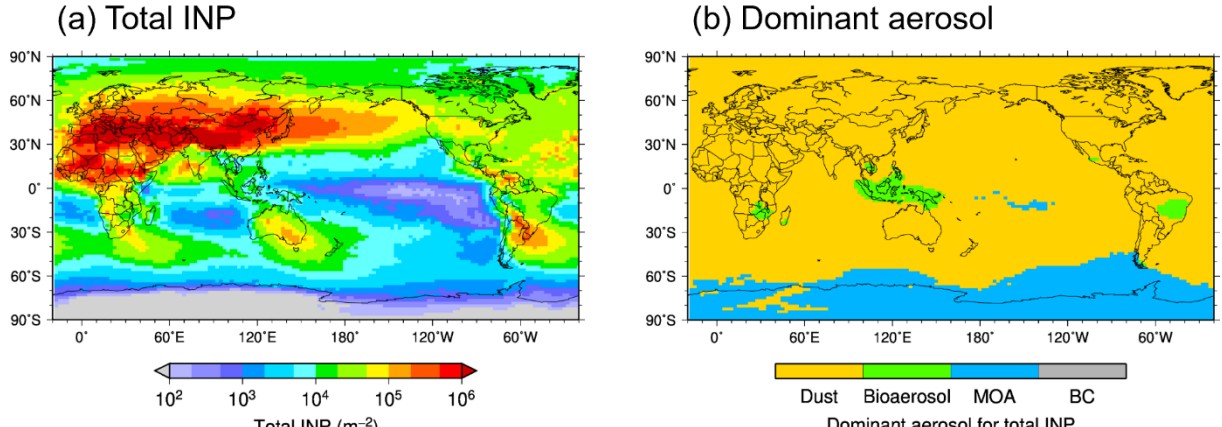

**Figure 5.** (a) Annual mean vertically integrated INP number concentrations of all aerosol species (total INPs). (b) Dominant aerosol species contributing the largest fraction of the total INP number concentration in each grid.

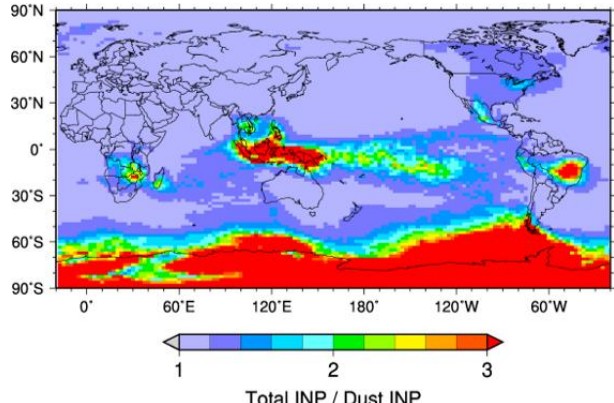


**Figure 6. Ratio of the annual mean vertically integrated INP number concentrations of all aerosol species (Fig. 5a) and dust (Fig. 4a).**





**Figure 7.** (a, c, e) Annual mean vertically integrated total INP number concentrations and (b, d, f) dominant aerosol species contributing the largest fraction of total INPs in (a, b) the upper, (c, d) middle, and (e, f) lower troposphere (same pressure levels as Fig. 3).



### 3.4 CREs via INPs

The CREs of dust and bioaerosol INPs are estimated from the differences between the 10-fold and 1-fold INP simulations
(Sect. 2.4) (the Base simulation in Table 2). Both dust and bioaerosols have positive shortwave CREs via INPs (+0.074 and
+0.027 W m$^{-2}$, respectively). These positive values likely result from two main factors: (1) because INPs increase the number
of ice crystals (with larger diameters) and reduce the number of water droplets (with smaller diameters) through the Wegener–
Bergeron–Findeisen process (Table S2), they lower the cloud albedo; and (2) because the increase in the number of ice crystals
is smaller than the decrease in the number of water droplets because of precipitation, there is an overall decrease in cloud
optical thickness. The shortwave CRE of dust via INPs is approximately 2.8 times that of bioaerosols. The longwave CRE of
dust via INPs is also positive (+0.16 W m$^{-2}$), likely because the increased number of ice crystals at high altitudes emit less
longwave radiation. In contrast, the longwave CRE of bioaerosols is negligible (−0.015 W m$^{-2}$). Consequently, the net
(shortwave + longwave) CRE of dust via INPs (+0.23 W m$^{-2}$) is about 19 times greater than that of bioaerosols (+0.012 W
m$^{-2}$).

Although the CREs of bioaerosol-derived INPs are small and subject to substantial uncertainty, the ratio of the CREs of
bioaerosols to those of dust via INPs (36 %, −9.4 %, and 5.2 % for shortwave, longwave, and net CREs, respectively; Table
2) is considerably higher than the global mean ratio of bioaerosols to dust INP number concentrations (2.5 %; Fig. 3a). This
discrepancy may be attributed to differences in the sensitivity of CREs to INP number concentration in regions where either
bioaerosols or dust are dominant. The sensitivity appears to be higher for bioaerosols than for dust, suggesting that even a
small number of bioaerosol-derived INPs might efficiently form ice crystals. This difference of sensitivity is likely due to the
fact that mixed-phase clouds at temperatures where bioaerosols dominate INP activity (warmer than approximately −10 °C)
tend to contain fewer total INPs, more water droplets, and fewer ice crystals.

    The CRE of all aerosol species via INPs in the Base simulation (0.24 W m$^{-2}$) is approximately equal to the sum of the
CREs of dust and bioaerosols via INPs (Table 2). This result highlights the crucial roles of dust and bioaerosols in determining
the radiative effects of aerosol-cloud interactions via INPs in mixed-phase clouds, with dust making a markedly larger
contribution than bioaerosols.

    We also estimate the CREs of all aerosol species via INPs when the ice nucleating ability of bacteria is constrained by the
year-round INP observations in Tokyo (Sect. 2.3). The observationally constrained simulation reasonably reproduces the
temperature dependence of the INP number concentrations observed in Tokyo in each month (Fig. S8) and the global INP
observations in China, Canada, and the Arctic, although it overestimates the INP number concentration observed in France by
an order of magnitude (Fig. S9). The shortwave and longwave CREs of all INP sources estimated by the observationally
constrained simulation are +0.051 and +0.59 W m$^{-2}$, respectively (Table 2). As a result, the estimated net CRE (+0.64 W m$^{-2}$)
is 2.7 times that of the Base simulation (+0.24 W m$^{-2}$). The net CREs scaled to the 1-fold − 0-fold differences (Sect. 2.3) are
estimated to be +0.071 W m$^{-2}$ for the Base simulation and +0.19 W m$^{-2}$ for the observationally constrained simulation (Table
2). Although we constrain the ice nucleating ability of bacteria to the INP observations in Tokyo, the differences in CREs



between the Base and observationally constrained simulations may reflect uncertainties in various factors, including not only the emission and ice nucleating ability of bacteria, but also INPs from other aerosol species and unknown sources. Further investigations of these factors are required to reduce uncertainties in estimation of INPs and their radiative effects through aerosol-cloud interactions.


**Table 2.** Annual and global mean cloud radiative effects (CREs) of dust, bioaerosols, and all aerosol species via INPs at the top of the atmosphere (W m$^{-2}$) for the Base and observationally constrained simulations, estimated from the 10-fold − 1-fold simulations. The values represent the averages and standard deviations of annual mean CREs for 10 years. The scaled net CREs corresponding to the 1-fold − 0-fold INPs differences were estimated by fitting the simulated net CREs for 1–50-fold dust INPs relative to 0-fold dust INPs using a power-law function of $y = ax^b$ (Fig. S2) (Sect. 2.4).

| | Shortwave CRE (10× − 1×) | Longwave CRE (10× − 1×) | Net CRE (10× − 1×) | Scaled net CRE |
|---|---|---|---|---|
| | | Base simulation | | |
| Dust INPs | +0.074 ± 0.019 | +0.16 ± 0.01 | +0.23 ± 0.02 | — |
| Bioaerosol INPs | +0.027 ± 0.016 | −0.015 ± 0.011 | +0.012 ± 0.012 | — |
| All INP sources | +0.096 ± 0.022 | +0.14 ± 0.01 | +0.24 ± 0.02 | +0.071 |
| | | Observationally constrained simulation | | |
| All INP sources | +0.051 ± 0.023 | +0.59 ± 0.02 | +0.64 ± 0.02 | +0.19 |

## 4 Conclusions

This study evaluated the number concentrations, spatial distributions, and CREs of INPs from multiple aerosol species (dust, bioaerosols, MOA, and BC) using the global climate-aerosol model CAM-ATRAS. By incorporating bioaerosols and MOA

as INP sources, the model reproduces INP observations worldwide more accurately than a simulation considering only dust as INPs. The global mean INP number concentration is dominated by dust (97 %) because of its larger atmospheric mass burden. However, bioaerosols and MOA are the major contributors in specific regions: bioaerosols dominate in the middle troposphere at low latitudes and in the lower troposphere at mid-latitudes in the Northern Hemisphere, whereas MOA dominates in the middle and lower troposphere at mid- and high latitudes in the Southern Hemisphere. The inclusion of bioaerosols and MOA



increases INP number concentrations by more than threefold in Antarctica, the Southern Ocean, Southeast Asia, South Africa, and Brazil, underscoring the regional importance of bioaerosols and MOAs in aerosol-cloud interactions via INPs.

The global mean net (shortwave + longwave) CREs $(10\times - 1\times)$ via INPs from dust and bioaerosols are +0.23 and +0.012 W m$^{-2}$, respectively. The sensitivity of CREs to INP number concentrations (i.e., CRE per unit INP number) appears to be higher for bioaerosols than for dust, likely because mixed-phase clouds at temperatures where bioaerosols dominate INP
activity tend to contain fewer total INPs, more water droplets, and fewer ice crystals. When the temperature dependence of INP activity is constrained using year-round INP observations in Tokyo, the scaled net CRE $(1\times - 0\times)$ via INPs from all aerosol species increases from +0.071 W m$^{-2}$ in the Base simulation to +0.19 W m$^{-2}$ in the observation-constrained simulation. These results highlight the importance of including INPs from multiple aerosol species and representing realistic INP number concentrations in model simulations to better understand aerosol-cloud interactions. Long-term global observations of aerosols
and INPs are essential for constraining climate models and reducing uncertainties in the estimation of INPs and their radiative effects.

In this study, we assumed that a small fraction of atmospheric bacteria—specifically *Pseudomonas syringae*—acts as efficient INPs, based on a previous study (Hummel et al., 2018). However, future research should aim to estimate the global surface distribution of such bacterial species using observational data and incorporate this distribution into models to improve
emission flux estimates. Considering the interannual variability of bioaerosol and MOA emissions is essential for assessing long-term changes in total INP concentrations and the CREs of aerosol-cloud interactions via INPs.

**Code and data availability**

Data used in this study are available upon request from the corresponding authors (KK and HM).

**Author contribution**

KK and HM conceived and designed the research. KK and ZR performed model simulations and data analysis. KK wrote the manuscript. KK and HM interpreted data, discussed their implications, and contributed to the manuscript.

**Competing interests**

The authors declare that they have no conflict of interest.

**Acknowledgements**

We thank Yutaka Tobo (National Institute of Polar Research, Japan) for providing INP observation data. Model simulations were performed using the supercomputer system of Osaka University, Japan.



**Financial support**

This research was supported by the Ministry of Education, Culture, Sports, Science, and Technology (MEXT) and the Japan Society for the Promotion of Science (JSPS) through KAKENHI Grants (JP22H03722, JP23H00515, JP23H00523,

JP23K18519, JP23K24976, and JP24H02225); the MEXT Arctic Challenge for Sustainability II (ArCS II; JPMXD1420318865) and 3 (ArCS-3; JPMXD1720251001) Projects; the Environment Research and Technology Development Fund 2-2301 (JPMEERF20232001) of the Environmental Restoration and Conservation Agency; the Joint Research Program of the Arid Land Research Center, Tottori University (05C2001); and National Institute of Polar Research (NIPR) through Special Collaboration Project (B25-02). This work was partly achieved through the use of SQUID at the D3 Center, Osaka

University.

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
