# Peer review of "Global Modeling of Ice Nucleating Particles of Multiple Aerosol Species and Associated Cloud Radiative Effects"

_EGUsphere, 2025_

## Referee Comment (RC1)

ACP review egusphere-2025-5850

Title: Global Modeling of Ice Nucleating Particles of Multiple Aerosol Species and Associated Cloud Radiative Effects

Author(s): Kei Kawai, Zhaoyi Ren, and Hitoshi Matsui

MS No.: egusphere-2025-5850

MS type: Research article

Iteration: Initial submission

General Comments:

This paper appears to be the first global modeling study of INPs where the cloud radiative effects (CREs) of respective INP species are estimated. The methods used appear to be state-of-the-science and the results conform to physical intuition. The paper is well written and well organized. The paper could be improved by adding a discussion of how an INP is defined, considering the temperature-dependent nucleating efficiency of an INP species. Some other ways to improve this paper are listed below. I recommend this paper for publication in ACP with minor revisions.

Major Comments:

Lines 21 – 26: Are there studies showing how increasing the ice fraction decreases cloud lifetime and cloud fraction due to the higher fall speeds of ice particles? This will affect the CRE. If so, this process with references should be mentioned. Relevant references might be Mitchell et al. (2008, GRL) and Eidhammer et al. (2017, J. Climate, p. 618).

Lines 41 – 43: Righi et al. (2025, ACP) may be relevant here since they show BC from aviation is not a significant INP.

Table 1: Please add median particle size to Table 1. This will make it clearer that the greater mass concentration of dust translates to a higher INP number concentration. (For example, anomalously large INPs may dominate the mass concentration but not the number concentration.)

Lines 89 – 91: Does this imply that the INP number concentration = the ice crystal number concentration (not the ice particle number concentration that is affected by aggregation)?

More specifically, how are INPs defined?  Do all INPs form ice crystals under favorable conditions?  Or is there a probability, like 10%, that an INP will form an ice crystal under favorable conditions?  If the former, then INP conc. = aerosol species # conc. x probability?  In either case, please clearly define INP.  It appears that Table 1 provides references for the INP probabilities used in this study.  Please discuss how these probabilities are used.

Lines 110 – 113: Roughly, what are the highest latitudes sampled by MODIS?  This may inform the reader which latitudes have the highest confidence.

Lines 119 – 121: Righi et al. (2025, ACP) also found that aviation soot has no significant impact on the INP concentration.  Consider adding this reference above to help justify your practice of ignoring anthropogenic BC as an INP source.

Lines 132 – 135: Ice nucleating ability tends to decrease with increasing temperature, but here it abruptly increases moving from -9 C to -5 C.  Was there a typo or is there a physical reason for this?  If the latter, please provide the reason.

Lines 204 – 208:  Same comment as for Lines 89 – 91.

**Technical Comments:**

Figure S8 in the Supplement:  Panels marked Jan 2016 through Jul 2016 should be dated as 2017.